# Conjugates of Chloramphenicol Amine and Berberine as Antimicrobial Agents

**DOI:** 10.3390/antibiotics12010015

**Published:** 2022-12-22

**Authors:** Julia A. Pavlova, Andrey G. Tereshchenkov, Pavel A. Nazarov, Dmitrii A. Lukianov, Dmitry A. Skvortsov, Vladimir I. Polshakov, Byasilya F. Vasilieva, Olga V. Efremenkova, Mikhail Y. Kaiumov, Alena Paleskava, Andrey L. Konevega, Olga A. Dontsova, Ilya A. Osterman, Alexey A. Bogdanov, Natalia V. Sumbatyan

**Affiliations:** 1Department of Chemistry, Lomonosov Moscow State University, 1/3 Leninskie Gory, 119991 Moscow, Russia; 2Center of Life Sciences, Skolkovo Institute of Science and Technology, 143028 Skolkovo, Russia; 3A.N. Belozersky Institute of Physico-Chemical Biology, Lomonosov Moscow State University, 1/40 Leninskie Gory, 119992 Moscow, Russia; 4Faculty of Fundamental Medicine, Lomonosov Moscow State University, 27/1 Lomonosvsky Ave., 119991 Moscow, Russia; 5Gause Institute of New Antibiotics, 11 B. Pirogovskaya Street, 119021 Moscow, Russia; 6Petersburg Nuclear Physics Institute, NRC “Kurchatov Institute”, 188300 Gatchina, Russia; 7Institute of Biomedical Systems and Biotechnologies, Peter the Great St. Petersburg Polytechnic University, 195251 Saint Petersburg, Russia; 8NRC “Kurchatov Institute”, 123182 Moscow, Russia; 9Shemyakin-Ovchinnikov Institute of Bioorganic Chemistry, 117997 Moscow, Russia

**Keywords:** chloramphenicol, berberine, C-13 derivatives of berberine, bacterial ribosome, bacterial membrane potential, antibiotic activity, biofilms

## Abstract

In order to obtain antimicrobial compounds with improved properties, new conjugates comprising two different biologically active agents within a single chimeric molecule based on chloramphenicol (CHL) and a hydrophobic cation were synthesized and studied. Chloramphenicol amine (CAM), derived from the ribosome-targeting antibiotic CHL, and the plant isoquinoline alkaloid berberine (BER) are connected by alkyl linkers of different lengths in structures of these conjugates. Using competition binding, double reporter system, and toeprinting assays, we showed that synthesized CAM-Cn-BER compounds bound to the bacterial ribosome and inhibited protein synthesis like the parent CHL. The mechanism of action of CAM-C5-BER and CAM-C8-BER on the process of bacterial translations was similar to CHL. Experiments with bacteria demonstrated that CAM-Cn-BERs suppressed the growth of laboratory strains of CHL and macrolides-resistant bacteria. CAM-C8-BER acted against mycobacteria and more selectively inhibited the growth of Gram-positive bacteria than the parent CHL and the berberine derivative lacking the CAM moiety (CH_3_-C8-BER). Using a potential-sensitive fluorescent probe, we found that CAM-C8-BER significantly reduced the membrane potential in *B. subtilis* cells. Crystal violet assays were used to demonstrate the absence of induction of biofilm formation under the action of CAM-C8-BER on *E. coli* bacteria. Thus, we showed that CAM-C8-BER could act both on the ribosome and on the cell membrane of bacteria, with the alkylated berberine fragment of the compound making a significant contribution to the inhibitory effect on bacterial growth. Moreover, we showed that CAM-Cn-BERs did not inhibit eukaryotic translation in vitro and were non-toxic for eukaryotic cells.

## 1. Introduction

Due to the increasing resistance of bacteria to antibiotics used in medical practice, along with the study of the mechanisms of the resistance and the interaction of antibiotics with their targets, it is important to develop new antimicrobial compounds, including the search for new natural antibiotics and the modification of already known molecules [1,2]. 

One of the promising approaches to the modification of antibiotics is the combination of two biologically active compounds or pharmacophores into one molecule that provides a new molecule with dual or improved action [3]. When creating new compounds using this approach, information about the targets of each of the components in the cell and the mechanisms of their action is taken into account. The molecular target of a large proportion of the known antibiotics is the bacterial ribosome, and as a result, many mechanisms of development of bacterial resistance are associated with mutations in the ribosome RNA and proteins [2,4]. When modifying ribosomal antibiotics, the main task, along with imparting new properties that lead to an expansion or narrowing of the spectrum of action of antimicrobial compounds, a decrease in toxicity, or an improvement in their bioavailability, pharmacokinetic and other properties, is to obtain compounds capable of acting against resistant strains [5,6].

The well-known and oldest antibiotic whose mechanism is associated with interaction with the bacterial ribosome and the arrest of bacterial protein synthesis is chloramphenicol (CHL). It binds to the peptidyl transferase center (PTC) of the bacterial ribosome [7] and inhibits peptide bond formation [8]. Despite the fact that this bacteriostatic antibiotic [9] is currently very limited in practice due to its side effects, the action that promotes the formation of biofilms [10,11], and the appearance of a large number of CHL-resistant strains, research on the modification of the chloramphenicol molecule continues nowadays [12,13]. This interest has been greatly facilitated by the clarification of the mechanism of action of CHL in the translation process [14,15,16]. Many chloramphenicol derivatives have been obtained by replacing the dichloroacetyl residue of CHL with some other moieties. Such a modification can be easily achieved by the chemical hydrolysis of CHL to chloramphenicol amine (CAM), followed by its conjugation with another molecule. CAM is practically inactive against bacteria [17] and does not bind to bacterial ribosomes [18]. However, modification of the amino group with acyl (and some other) residues allows resulting compounds to restore the ability to bind to the ribosome at the CHL binding site and inhibit the biosynthesis of bacterial proteins [13,19,20,21]. It has been shown that the insertion of charged groups in the structure of CHL can improve the affinity of derivatives to the ribosome [20,22,23], and in some cases, leads to the production of analogs acting against resistant bacterial strains [20,23]. The presence of a hydrophobic cation in a molecule contributes to a higher affinity of such compounds to the ribosome and can change the specificity of this interaction, as has been shown for CAM and triphenylphosphonium (TPP) conjugates (CAM-Cn-TPP) [24]. CAM-Cn-TPP can inhibit bacterial growth both by acting on bacterial ribosomes and by reducing the membrane potential of bacteria. However, these compounds turned out to be rather toxic to eukaryotic cells [23].

An isoquinoline plant alkaloid berberine (BER) also has been attributed to hydrophobic penetrating cations that accumulate in cells driven by the membrane potential [25]. The structure of BER contains condensed aromatic rings, and the positive charge of the quaternary nitrogen atom is delocalized over a conjugated system of electrons (Figure 1). The positive charge and the planar structure of the molecule contribute to the interaction of BER with nucleic acids, intercalation, and stacking [26,27,28]. BER can act as an iron chelator [29], as well as interact with anionic and nucleophilic moieties of various biomolecules and interfere with multiple cellular signaling pathways [30,31]. Due to structural and chemical properties, BER and other alkaloids of its group exhibit an exceptionally wide range of biological activity [32,33,34,35]; BER has anti-inflammatory [30,34], hepatoprotective [36], antihyperlipidemic [37], antidiabetic [38], antiviral [39], antiproliferative [33,34,35], antibacterial [25,40,41,42,43] and many other pharmacological effects [27]. Plant extracts containing berberine alkaloids are widely used in traditional medicine in a number of countries to treat various infections [44,45].

It has been reported that the antibacterial effect of BER may be associated with membrane damage [25,28], in particular, with a decrease in membrane potential [25]. As a result of the binding of BER to DNA and RNA, their structure can change, and therefore BER can influence the processes of replication and transcription and, thus, indirectly disrupts protein biosynthesis [26,27,28]. It has also been shown that BER and related compounds can interact with proteins [28] and affect the processes of replication and transcription or bacterial cell division by inhibiting the corresponding enzymes (e.g., topoisomerases [46]) or protein FtsZ [47,48]. 

Although BER has antimicrobial activity, relatively large doses are required for its action against the vast majority of bacterial strains [40]. Such low antibacterial activity of BER is explained by the fact that BER is extruded from bacterial cells by efflux pumps (e.g., multidrug-resistance pump NorA in *St. aureus* cells) [49]. The multidrug resistance (MDR) pump complex of any bacterium is a nonspecific defense system that removes xenobiotics from the bacterial cell and protects the bacteria from the hostile environment [50]. It is not surprising that in plant extracts containing BER, there are, in addition, efflux pump inhibitors, in particular, 5′-methoxyhydnocarpin, which strongly potentiates the antibacterial effect of BER [49,51,52].

Among the large number of derivatives of BER and related compounds, many are effective antibacterial agents [46,48,49,53,54,55,56,57,58,59,60,61,62]. It has been shown that the most suitable positions for modification in the berberine molecule to obtain antibacterial compounds with improved properties are positions C-9 [46,49,52,55,56,57] and C-13 [48,54,55] or both at the same time [54]. The introduction of various aromatic substituents into position 13 of the berberine molecule [48,54], as well as substituted 13-vinyl moieties [55,59] or the conjugation with an MDR pump inhibitor [61], leads to increased antibacterial activity and, in some cases, selectivity of the action of antimicrobial compounds. 

In the current study, we continued our research on the synthesis and exploration of antimicrobial compounds by combining two different biologically active agents within a single chimeric molecule based on chloramphenicol and a hydrophobic cation [23,24], with the goal of obtaining a new group of CHL derivatives with potentially improved or novel properties. To this end, the dichloromethyl group of the parent CHL compound was replaced with C13-modified berberine residue containing an alkyl chain whose length varied from one to 8 methylene groups, resulting in CAM-Cn-BER molecules (Figure 1). Using various biochemical assays, we have shown possible mechanisms of action of synthesized CAM-Cn-BERs as antibacterial agents. We also tested the ability of CAM-Cn-BERs to suppress laboratory bacterial strains, including those resistant to CHL and macrolides, and to influence biofilm formation, as well as the toxicity of new conjugates for eukaryotic cells.

## 2. Results and Discussion

### 2.1. Synthesis of CAM-Cn-BER

Conjugates of CAM and BER were designed as analogs of CHL by replacing the dichloromethyl group of CHL with an alkylamide of berberine acid (13-carboxymethyl berberine), resulting in CAM-Cn-BER molecules (Figure 1). It was assumed that the antibacterial effect of these compounds would be due to both interaction with the bacterial ribosome (similarly to CHL) and their action on bacterial membranes (similarly to BER). Previously, it has been shown that when interacting with the bacterial ribosome, the amphenicol fragment of a number of CAM derivatives bound at the CHL site within the PTC, and the amide substituents were directed towards the exit from the nascent peptide exit tunnel (NPET) and formed various contacts with its elements [23,24]. 

According to our present idea, when CAM-Cn-BER is binding to bacterial ribosome, the amphenicol moiety would anchor the compound in the CHL binding site, and a berberine moiety would provide non-specific interactions with negatively charged phosphates of the 23S rRNA and stacking with nucleobases of NPET similarly to CAM-Cn-TPPs [24]. Moreover, the berberine group in these compounds, being a membrane-penetrating cation [25], should allow the CAM-Cn-BER to get inside bacterial cells. The variable length of the linker connecting the two terminal moieties of these compounds (CAM and BER) would ensure the optimal binding of CAM-Cn-BER to nucleotides of 23S rRNA in different NPET regions of the ribosome, as it has been observed for CAM-Cn-TPP [23,24]. The choice of the linker length was also based on the data on the biological activity of berberine analogs containing alkyl bridges [54,63]. CAM-Cn-BER (n = 1, 2, 3, 5, 8) were constructed so that the amino group of CAM was connected to the carboxymethyl group at the 13 position of BER through linkers formed by omega-amino acids. 

To synthesize CAM-Cn-BERs, a 13-carboxymethyl derivative of tetrahydroberberine (thBER-CH_2_-COOH) was obtained from BER according to the previously described procedures in three steps: BER was reduced with sodium borohydride yielding dihydroberberine, followed by enamine alkylation with methyl bromoacetate and immediate reduction of unstable 13-substituted iminium salt to give the 13-substituted tetrahydroberberine, which was hydrolyzed in alkaline conditions and gave thBER-CH_2_-COOH [63,64,65] (Figure 1). To obtain modified chloramphenicol amine (CAM-Cn-NH_2_), CHL was hydrolyzed under acidic conditions [17], and the resulting CAM was conjugated with hydroxysuccinimide ester of protected omega-amino acids (Boc-AA-OSu) followed by deprotection of amino group. Then the conjugation of thBER-CH_2_-COOH and CAM-Cn-NH_2_ was carried out to obtain CAM-Cn-thBER. The resulting compounds were further oxidized in the presence of *N*-bromosuccinimide (NBS) to obtain target compounds (CAM-Cn-BER) containing the native form of the berberine fragment. The general synthesis scheme is shown in Figure 1. The compounds were purified by silica gel column chromatography and analyzed by LC-MS and NMR. 

The structure of the CAM-C8-BER was unambiguously confirmed by two-dimensional NMR methods: ^1^H-^1^H DQF-COSY, ^1^H-^1^H ROESY, ^13^C-^1^H HSQC, ^13^C-^1^H HMBC (Appendix A).

### 2.2. Synthesis of CH_3_-Cn-BER

As controls for further biochemical and microbiological tests, *n*-hexylamide of berberine acid (CH_3_-C5-BER) and *n*-nonylamide of berberine acid (CH_3_-C8-BER) were obtained in this work. The synthesis of CH_3_-Cn-BERs was performed by the acylation of *n*-hexylamine and *n*-nonylamine with a 13-carboxyl derivative of tetrahydroberberine, thBER-CH_2_-COOH, using *N*,*N′*-dicyclohexylcarbodiimide (DCC) according to the scheme shown in the Appendix A. Oxidation of the obtained compounds (CH_3_-Cn-thBER) was carried out similarly to the procedure used for CAM-Cn-BERs using NBS. 

### 2.3. Preliminary Assessment of the Antibacterial Activity and Mechanism of Action of CAM-Cn-BER in the Double Reporter System pDualrep2

The JW5503 (*ΔtolC*) (KanS) pDualrep2 strain was used for a preliminary assessment of the mechanism of the antibacterial action of the obtained compounds [66]. It is an *E. coli ΔtolC* reporter system based on two proteins, RFP and Katushka2S, whose fluorescence can be independently detected. The described system can be used for screening inhibitors targeting either protein synthesis or DNA replication.

In the pDualrep2 reporter system, the expression of far-red fluorescent protein, Katushka2S, occurs only in the presence of ribosome-stalling compounds because the gene of this protein is inserted downstream of the genetically modified tryptophan attenuator. The expression of red fluorescent protein, RFP, can be induced by compounds that trigger SOS response because the gene *rfp* is placed under the control of the SOS-inducible *sulA* promoter. The fluorescence of these proteins can be easily detected by various fluorescence scanners. The resulting image is an overlay of two images of independent RFP (Cy3 channel) and Katushka2S (Cy5 channel) scans. RFP fluorescence is shown as green pseudocolor (Figure 2A, LEV, green pseudocolor ring) and Katushka2S fluorescence as red pseudocolor (Figure 2A, CHL, red pseudocolor ring). 

For CAM-C5-BER and CAM-C8-BER, as well as for the positive control, CHL, red pseudocolor rings are observed due to the expression of Katushka2S, indicating that these compounds specifically inhibit protein synthesis as previously described CAM-Cn-TPPs [23,24]. At the same time, the size of the zone of inhibition by CAM-C8-BER is larger than in the presence of CAM-C5-BER, which indicates its better efficiency compared with the analog containing a shorter linker. CAM-C1-BER and CAM-C2-BER do not cause zones of inhibition, CAM-C3-BER slightly inhibits bacterial cell growth (dark area in the middle) but does not induce either of the two reporters. BER causes a small zone of inhibition (Appendix A) but does not induce the described reporters, which suggests that it has antibacterial properties, but the mechanism is different from those that can be detected by the pDualrep2 reporter system.

Thus, testing of the compounds using a double reporter system showed that the conjugates of CHL and BER (CAM-C5-BER and CAM-C8-BER) have antibacterial activity against *E. coli*
*ΔtolC* strain, and their possible mechanism of action is similar to the one of the parent compounds—CHL.

### 2.4. CAM-Cn-BER and CH_3_-Cn-BER Exhibit Antibacterial Activity against Various Strains, including Resistant

The obtained conjugates of CHL and BER, which, according to the reporter system, have a mechanism of action similar to CHL, were further tested on strains resistant to the parent compound. One of the most common mechanisms of resistance to CHL is the acetylation of its methylene hydroxyl group by chloramphenicol acetyltransferases [67]. Therefore, the *E. coli*
*ΔtolC*-CAT and *B. subtilis*-CAT strains transformed with the plasmids carrying the *cat* gene encoding chloramphenicol acetyltransferase were chosen for the study. Figure 2B shows that CHL gives a barely visible zone of inhibition, and the tested substances have significant antibacterial activity against *E. coli*
*ΔtolC*-CAT strain, especially CAM-C5-BER and CAM-C8-BER.

The binding site of CHL on ribosome partially overlaps with the binding site of macrolides [68]; moreover, this applies to CAM-Cn-BER since instead of a dichloroacetyl residue of CHL, they contain bulky alkylamide-berberine substituent directed towards the exit from NPET. One of the common mechanisms of resistance to macrolides is the substitution A2058G in the 23S rRNA. The amino group of guanosine sterically prevents the binding of the lactone ring of macrolides, resulting in resistance [69]. Therefore, the obtained conjugates of BER and CHL were tested on *E. coli* SQ110 *ΔtolC* (A2058G) strain resistant to macrolides (Figure 2C). Erythromycin (ERY), a member of macrolide antibiotics, has no inhibitory activity, while the substances tested have variable effectiveness. CAM-C5-BER and CAM-C8-BER have the best activity among the obtained substances.

For the substance CAM-C8-BER, which showed the best antibacterial activity, minimum inhibitory concentrations (MIC) were determined against *E. coli*
*ΔtolC* and *B. subtilis* strains, as well as against CHL-resistant *E. coli*
*ΔtolC*-CAT and *B. subtilis*-CAT strains (Table 1). It can be noted that CAM-C8-BER and CH_3_-C8-BER act more effectively on a strain deficient in TolC-containing pumps, which indicates the involvement of TolC-containing pumps in the removal of substances from the cell, as was shown for CHL [70]. 

Notably, the tested substances are active against CHL-resistant strains *E. coli*
*ΔtolC*-CAT and *B. subtilis*-CAT, indicating that they can overcome resistance induced by the activity of chloramphenicol acetyltransferases. However, the control substance CH_3_-C8-BER also shows significant antibacterial activity against both resistant and non-resistant strains. Apparently, the berberine fragment in the conjugate mainly contributes to the effect of CAM-C8-BER on resistant strains. 

The activity of the obtained substances was also studied against various Gram-positive and Gram-negative bacteria, as well as against mycobacteria and fungi (Appendix A). Unlike CHL, which is active against all studied strains except fungi, and CH_3_-C8-BER, which inhibits the growth of Gram-positive strains of bacteria, mycobacteria, fungi, and yeasts, CAM-C8-BER exhibits a narrow spectrum of activity (Table 2). CAM-C8-BER exhibits inhibitory activity against mycobacteria and *L. mesenteroides*, a vancomycin-resistant strain that causes rare but difficult-to-treat infectious diseases [71,72]. BER at the concentration under study was active only against strains of mycobacteria; it did not show activity against other strains. 

### 2.5. CAM-Cn-BER Selectively Inhibit Prokaryotic Translation, Allowing the Formation of Short Peptides, and Bind to the Bacterial Ribosome

According to the results of testing the compounds using the pDualrep2 reporter system, the antibacterial activity of CAM-Cn-BERs is associated with the inhibition of protein biosynthesis. Therefore, the ability of CAM-Cn-BERs to inhibit protein biosynthesis as well as binding to bacterial ribosomes was studied (Figure 3).

The ability of substances to inhibit protein biosynthesis was tested using the firefly luciferase reaction in a cell-free transcription-translation system based on the S30 extract of *E. coli*. CAM-C8-BER and CAM-C5-BER inhibit bacterial translation of luciferase mRNA similarly to CHL (Figure 3A). At the same time, both CAM-C8-BER and CAM-C5-BER had no effect on eukaryotic in vitro translation (Appendix A), which was revealed by a similar approach using the eukaryotic in vitro translation system. Thus, the studied substances selectively inhibit prokaryotic translation but do not affect eukaryotic translation, which is important for the possibility of using these compounds as antibacterial drugs.

The affinity of CAM-Cn-BERs for the bacterial 70S ribosome (Figure 3B) was assessed by a competition-binding assay using BODIPY-labeled erythromycin (BODIPY-ERY) [22,73,74]. This fluorescent derivative was chosen because of the overlap of the binding sites for CHL and ERY in the ribosome [69]. According to the results, the conjugation of the berberine fragment with CAM improved the affinity of the compounds (CAM-Cn-BER) to the ribosome compared to the parent CHL, similar to that as it has been shown earlier for amino acid, peptide, and triphenylphosphonium analogs of CHL [18,22,23,24]. However, there is no clear correlation between the affinity of the compounds for the ribosome and the length of the linker connecting the chloramphenicol and berberine fragments, which may indicate that the berberine fragment does not have one specific binding site, and, most likely, can form a stacking interaction with various nucleotides of the ribosomal tunnel. 

The highest affinity for the ribosome is observed for CAM-C8-BER as well as for CAM-C3-BER (K_Dapp_ = 0.16 ± 0.02 µM and 0.13 ± 0.02 µM respectively). It is worth noting that if CAM-C8-BER turned out to be the most active in tests on bacteria, then CAM-C3-BER did not show any activity. Obviously, the ability to bind to the ribosome in the PTC or NPET region is not a sufficient condition for the antibacterial action of the compounds [22], and their capability to penetrate the permeability barriers of the bacterial cell is of decisive importance. At the same time, alkylamide-berberines (CH_3_-Cn-BER) practically do not displace the fluorescent ligand from its binding site (Appendix A) but exhibit antibacterial properties, which, apparently, are not associated with an effect on the translation process.

### 2.6. CAM-Cn-BER (n = 5, 8) Act according to the Context Specificity of Chloramphenicol 

CHL is a classic PTC-acting antibiotic that has a context-specific mechanism of action. It has been shown that CHL inhibits protein biosynthesis most effectively when the ribosome carries a nascent peptide with alanine, serine, or threonine in the penultimate position [14,15]. According to structural data, the penultimate residue of the nascent peptide directly influences antibiotic affinity to the ribosome [16]. The CHL-bound ribosome can still catalyze the formation of a peptide bond if aminoacyl-tRNA carries glycine [75]. Thus, the action of CHL depends on the sequence of the template from which the protein is synthesized.

The mechanism of action of CAM-Cn-BERs during the translation process was investigated using a primer extension inhibition assay (toeprinting). For this experiment, *RST-2* mRNA was chosen as a template. On the matrix used, CHL causes ribosome arrest at the fifth codon (isoleucine), while alanine occupies the penultimate position of the nascent peptide (Figure 3C). CAM-C3-BER causes a weak stop at the start codon and at the fifth. CAM-C5-BER and CAM-C8-BER cause ribosome arrest mainly at the fifth codon, similarly to CHL, and with less intensity at the seventh (leucine), while the penultimate residue in the nascent peptide corresponds to threonine.

The results show that on the *RST-2* template, CAM-Cn-BER (n = 5, 8) acts according to the context specificity of CHL, allowing the formation of short peptides. So, CAM-Cn-BERs act differently from previously obtained amino acid CHL derivatives [22] and especially CAM-Cn-TPPs [23], which have unique context-specificity that differs from that of the original CHL.

### 2.7. Molecular Docking of CAM-Cn-BER Suggests Possible Explanation of Toeprinting Results

In order to explain the difference in the context specificity of CAM-Cn-BERs, we performed molecular docking calculations under the conditions used in the toeprinting assay. For this, in the crystal structure of the 70S *T. thermophilus* ribosome in complex with aminoacylated A-site Phe-NH-tRNA^Phe^, peptidyl P-site fMTHSMRC-NH-tRNA^Met^, and deacylated E-site tRNA^Phe^ (PDB ID: 8CVL [76]), aa-tRNA was deacylated, and the peptide fMTHSMRC was replaced by amino acid sequence fMKFAI, that causes translation arrest in the presence of CHL, CAM-C5-BER, and CAM-C8-BER (Section 2.6 and Figure 3C). CAM-Cn-BERs were docked into the resulting structure, in which the peptide of P-site fMKFAI-NH-tRNA^Met^ was considered a flexible part of the receptor during calculations.

As a result of molecular docking, the structures of the ribosome/fMKFAI-NH-tRNA^Met^/CAM-Cn-BER (n = 1, 2, 3, 5, 8) ternary complexes were obtained. It was found that for the molecules with short linkers (n = 1, 2), no conformations were found, in which the chloramphenicol part of the molecules is located at the binding site of the parent antibiotic CHL. These data are in good agreement with toeprinting results, in which there is practically no translation arrest for the corresponding compounds in the presence of the synthesized pentapeptide fMKFAI (Figure 3C, lanes 5, 6). This is probably due to the fact that the BER residue with short linkers does not fit at the very beginning of the NPET, along with the growing peptide.

For CAM-Cn-BER with longer linkers (n = 3, 5, 8), conformations similar to that of CHL were obtained. In the resulting structures, the nitrophenyl group of CAM-Cn-BERs is located between the nucleotides A2451 and C2452 and forms a stacking interaction with them (Figure 4A,B), which is characteristic of both CHL [77] and its analogs [22,24]. In the case of CAM-C5-BER and CAM-C8-BER conformations, the CHL fragment of the molecules and the Ala4-Ile5 residues are only slightly shifted compared to the X-ray crystal structure of CHL and P-site MAI-tripeptidyl-tRNA analog, MAI-ACCA, in complex with the *T. thermophilus* ribosome (PDB ID: 7RQE [16]) (Figure 4C,D). The berberine fragment of the compounds is directed inside the NPET and can form various interactions with 23S rRNA nucleotides, including A2058, A2062, m^2^A2503, and C2586 (Figure 4A,B), which play an important role in binding macrolides and other antibiotics [69,78], and also forms π-π stacking with the benzene ring of Phe3 residue of the fMKFAI peptide (Figure 4C,D). 

The results of molecular docking are consistent with the results obtained for CAM-Cn-BERs by the toeprinting method and demonstrate the possibility of fMKFAI peptide formation during biosynthesis on the ribosome in the presence of CAM-C5-BER and CAM-C8-BER.

### 2.8. CAM-C8-BER Causes a Decrease in the Membrane Potential of B. subtilis

It is known that BER penetrates bacterial membranes in a charged form and electrophoretically accumulates inside *St. aureus* cells [25]. It has also been shown that some berberine derivatives can disrupt mitochondrial membrane potential [79].

Previously obtained CAM-Cn-TPPs, as well as alkyl-TPPs, showed the ability to reduce the membrane potential of bacteria [23]. Thus, we further examined the effect of CAM-Cn-BERs in comparison with alkylamide-berberines (CH_3_-Cn-BER) on the bacterial membrane potential of *B. subtilis*. Gramicidin A, a channel-forming antibiotic that causes the disappearance of the bacterial membrane potential, was used as a control. The change in the membrane potential of *B. subtilis* in the presence of the studied compounds was assessed by measuring the fluorescence of the potential-sensitive dye diS-C_3_-(5). This dye accumulates inside the cells under the action of the potential, which causes a decrease in its fluorescence. The addition of substances that reduce the membrane potential leads to the release of the dye and an increase in fluorescence.

CAM-C8-BER rapidly reduces the membrane potential to a value observed for the channel-forming antibiotic gramicidin A at a concentration of 1.5 µg/mL (2.01 µM) (Figure 5). The CH_3_-C8-BER shows a similar effect but at a slightly lower concentration of 0.75 µg/mL (1.44 µM). CAM-C5-BER and CH_3_-C5-BER do not cause some significant decrease in the membrane potential, even at ten times higher concentrations. 

Thus, the action of CAM-C8-BER can have a double effect on bacterial cells associated with the inhibition of protein biosynthesis (because of a CHL fragment in its structure [12,13]) and depolarization of the bacterial membrane (due to the presence of a BER moiety [25,28]). A similar double mechanism of action was previously described for CAM-Cn-TPPs [23].

### 2.9. CAM-C8-BER Does Not Have CHL Ability to Induce Biofilms 

The formation of biofilms is a protective mechanism of the bacterial population and determines the overall resistance [10,11]. This process is negative in terms of the development of modern drugs. CHL is known to be a bacteriostatic antibiotic that promotes biofilm formation [9]. On the contrary, BER is capable of suppressing the formation of biofilms in cultures of Gram-positive [80,81] and Gram-negative bacteria as well as mycobacteria [82], fungi [83], and pathogenic yeasts [84]. The mechanism of anti-biofilm action of BER was reported to associate with the modulation of bacteria quorum sensing regulation [85,86] and the inhibition of MDR pumps [87], which are actively involved in the formation of biofilms by exporting quorum sensing signals [50].

Crystal violet assays were used to evaluate the effect of CAM-C8-BER on the formation of *E. coli* biofilm compared to CHL. *E. coli* cells were exposed 20 h with compounds, and their cell density (OD_620_, back curves) and surface attachment (OD_595_, grey bars) were measured (Figure 6). The effect of the compounds on the biofilm formation process was assessed by comparing the change in the quantity of cells in a sessile and a planktonic population. The absence of induction of biofilm formation is observed for CAM-C8-BER at sub-inhibitory concentrations in contrast to CHL, which induces the redistribution of bacteria cells into an attached population. 

Thus, CAM-C8-BER does not destroy biofilms, like BER, but also does not induce their formation, unlike CHL. It is obvious that such an action of the conjugate is provided by the presence of the berberine moiety in the molecule, which causes the decrease in the membrane potential when acting on bacteria and, possibly, the inhibition of MDR pumps.

### 2.10. CAM-C8-BER and CAM-C5-BER Are Non-Toxic for Mammalian Cells

We have previously shown that synthesized substances do not affect the process of eukaryotic translation. To confirm the absence of cytotoxicity of CAM-Cn-BERs for mammalian cells, we used the Mosmann (“MTT”) assay [88]. 

The results show that the tested compounds (CAM-Cn-BER) are non-cytotoxic (Table 3), which is an important property for their further possible application. However, CH_3_-Cn-BER (n = 5, 8), as well as BER itself, which are parts of the molecular structure of CAM-Cn-BERs, exhibit varying degrees of toxicity in the studied cell lines, but they are less toxic than doxorubicin, which is a highly toxic drug used as a control. CH_3_-C8-BER has much greater toxicity to eukaryotic cell lines than CAM-C8-BER. It should also be noted that, unlike the previously described CAM-Cn-TPPs [23], both CAM-Cn-BERs and CH_3_-Cn-BERs do not show any selectivity against cancer cells (MCF7 and A549, Table 3) compared to the noncancerous cells (VA13 and HEK293T). 

## 3. Materials and Methods

### 3.1. Chemicals and Materials

The following reagents and solvents were used: berberine chloride (BER) (TCI, Tokyo, Japan); NaBH_4_ (Acros Organics, Geel, Belgium); CH_2_Cl_2_, dimethylformamide (DMF), pyridine, trifluoroacetic acid (TFA) (PanReac AppliChem, Darmstadt, Germany); methyl bromoacetate, *N*-bromosuccinimide (NBS), diisopropylethylamine (DIPEA), LiOH (Sigma-Aldrich, Steinheim, Germany); 1-hydroxysuccinimide (HOSu) (Sigma-Aldrich, Tokyo, Japan); chloramphenicol (CHL), erythromycin (ERY), levofloxacin (LEV) (Sigma-Aldrich, Shanghai, China); CHCl_3_, ethyl acetate, diethyl ether, acetic acid (glacial), toluene (Chimmed, Moscow, Russia); methanol (Merck, Darmstadt, Germany); hydrochloric acid (Irea 2000, Moscow, Russia); aqueous ammonia (Sigma Tech, Khimki, Russia); DCC, *n*-hexylamine (Fluka, Neu-Ulm, Germany); di-*tert*-butyl dicarbonate (Boc_2_O) (ABCR, Karlsruhe, Germany); 9-aminopelargonic acid (Reanal, Budapest, Hungary); Boc-6-aminocaproic acid (Chem-Impex, Wood Dale, IL, USA); crystal violet (Lenreactiv, St. Petersburg, Russia). *n*-Nonylamine hydrobromide was synthesized according to the procedure described in [89]. {[(1*R*,2*R*)-1,3-dihydroxy-1-(4-nitrophenyl)propan-2-yl]carbamoyl}methylammonium trifluoroacetate (CAM-C1-NH_2_·TFA), 2-{[(1*R*,2*R*)-1,3-dihydroxy-1-(4-nitrophenyl)propan-2-yl]carbamoyl}ethyl-1-ammonium trifluoroacetate (CAM-C2-NH_2_·TFA) [22], 3-{[(1*R*,2*R*)-1,3-dihydroxy-1-(4-nitrophenyl)propan-2-yl]carbamoyl}propyl-1-ammonium trifluoroacetate (CAM-C3-NH_2_·TFA) [23], and the fluorescent erythromycin derivative, BODIPY-ERY [90], were synthesized as described previously. 

### 3.2. Chromatography

TLC was performed on Kieselgel 60 F254 plates (Merck, Darmstadt, Germany); for column chromatography, Silica gel 60 (0.063–0.200 and 0.04–0.063 mm, Macherey-Nagel, Düren, Germany) was used. Compounds containing UV-absorbing groups were detected with a UV-cabinet Camag (Omicron Research, Hungerford, UK); substances with free or Boc-protected amino groups were visualized by a ninhydrin reagent. BER and its derivatives were detected using Dragendorff’s reagent.

### 3.3. Liquid Chromatography-Mass Spectrometry

Liquid chromatography-mass spectrometry was performed using a UPLC/MS/MS system consisting of an ACQUITY UPLC chromatograph from Waters (Milford, MA, USA) and a TQD quadrupole mass-spectrometer (Waters) with registration of positive ions using the ESI MS method with an ACQUITY BEH column C18 (1.7 microns, 50 × 2.1 mm, Waters), flow rate 0.5 mL/min, 35 °C, and elution with a gradient of 5–100% CH_3_CN in 20 mM of HCOOH for 4 min.

### 3.4. ^1^H and ^13^C NMR

^1^H and ^13^C NMR spectra were recorded with a Bruker Avance spectrometer (Bruker, Billerica, MA, USA) with operating frequencies 400 and 600 MHz for ^1^H, 101 and 151 MHz for ^13^C, at 298 K in DMSO-*d_6_* using tetramethylsilane as an internal reference. Assignment of ^1^H and ^13^C signals was carried out using one-dimensional ^1^H and ^13^C spectra and two-dimensional ^1^H-^1^H ROESY (320 ms mixing time), ^1^H-^1^H DQF-COSY, ^13^C-^1^H HSQC, and ^13^C-^1^H HMBC experiments. Spectra were processed by the NMRPipe software [91] using a standard protocol that includes the Lorentz-Gauss window, forward-backward linear prediction, and polynomial baseline correction. One-dimensional NMR spectra were processed and analyzed using Mnova software (Mestrelab Research, Santiago de Compostela, Spain). Two-dimensional spectra were analyzed with the NMRFAM-Sparky software [92]. For the compounds containing a tetrahydroberberine moiety, the NMR signals corresponding to different diastereomers are separated by an ampersand.

### 3.5. Synthetic Procedures

The scheme for the synthesis of CAM-Cn-BER (n = 1, 2, 3, 5, 8) is represented in Figure 1. CAM-Cn-NH_2_ was obtained in three stages from CHL. The first stage included the hydrolysis of CHL to CAM, according to [17]. In the second stage, conjugation of CAM and *N*-hydroxysuccinimide ester of omega-amino acids containing Boc-group at the amino group (Boc-AA-OSu) was carried out. In the third stage, the removal of the Boc-group was performed. [(13*RS*,13a*RS*)-9,10-dimethoxy-5,8,13,13a-tetrahydro-6*H*-benzo[*g*]-1,3-benzodioxolo[5,6-*a*]quinolizin-13-yl]-acetic acid (thBER-CH_2_-COOH) was obtained from BER according to [93]. See also Appendix A for more detailed information on procedures for the synthesis of CAM-C1-BER, CAM-C2-BER, CAM-C3-BER, CAM-C5-BER, CAM-C8-NH_2_, CH_3_-C5-BER, CH_3_-C8-BER.

*N-[(1R,2R)-1,3-dihydroxy-1-(4-nitrophenyl)propan-2-yl]-9-(2-[(13RS,13aRS)-9,10-dimethoxy-5,8,13,13a-tetrahydro-6H-benzo[g]-1,3-benzodioxolo[5,6-a]quinolizin-13-yl]acetamido)nonanamide (CAM-C8-thBER).* To the cold solution of 206 mg (0.475 mmol) of thBER-CH_2_-COOH·HCl and 77 mg (0.673 mmol) of HOSu in 10 mL of DMF, 139 mg (0.673 mmol) of DCC was added at 0 °C. The mixture was stirred for 2 h at 0 °C and overnighted at 4 °C. Then 247 mg (0.673 mmol) of CAM-C8-NH_2_ and 234 µL (1.346 mmol) of DIPEA were added, and the resulting mixture was stirred at RT overnight. Then the reaction mixture was diluted with a tenfold excess of water, and 1N aqueous HCl was added dropwise to pH 6. The mixture was then extracted with ethyl acetate (3 × 20 mL), and the combined organic extracts were washed with saturated NaCl solution (10 mL). The organic layer was dried over anhydrous Na_2_SO_4_, and the volatiles were evaporated in vacuo. The target product was isolated on a silica gel column eluting with a solvent system of CHCl_3_:MeOH = 9:1. As a result, a light yellow solid was obtained. Yield: 147 mg (41%); TLC: *R_f_* (CHCl_3_:MeOH, 9:1) 0.47; LC-MS *m*/*z* calculated for C_40_H_51_N_4_O_10_ (M + H)^+^: 747.36, found 747.47; *t*_R_ = 1.72 min; ^1^H NMR (DMSO-*d*_6_, 400 MHz) δ (ppm) 8.14 (2H, d, *J* = 8.6 Hz, *o*-H NO_2_-Phe), 7.58 (2H, d, *J* = 8.6 Hz, *m*-H NO_2_-Phe), 7.52 (1H, t, *J* = 5.6 Hz, -CH_2_-NH-CO-), 7.44 (1H, d, *J* = 9.1 Hz, -CH-NH-CO-), 6.85 (1H, d, *J* = 8.5 Hz, H-12), 6.83 (1H, s, H-1), 6.803 & 6.795 (1H, d, *J* = 8.5 Hz, H-11), 6.67 (1H, s, H-4), 5.953 (1H, d, *J* = 1.0 Hz, O-CH_2_^a^-O), 5.945 (1H, d, *J* = 1.0 Hz, O-CH_2_^b^-O), 5.84 (1H, br s, -CH-OH), 5.03 (1H, br s, -CH-OH), 4.86 (1H, t, *J* = 5.7 Hz, -CH_2_-OH), 4.06 (1H, d, *J* = 16.0 Hz, H-8^a^), 4.01 (1H, dddd, *J* = 9.1, 8.6, 5.9, 2.4 Hz, -CH-NH-CO-), 3.744 & 3.736 (3H, s, C_10_-OCH_3_), 3.72 (3H, s, C_9_-OCH_3_), 3.62 (1H, dt, *J* = 9.3, 2.9 Hz, H-13), 3.61 (1H, d, *J* = 2.9 Hz, H-13a), 3.54 (1H, ddd, *J* = 9.4, 8.4, 3.8 Hz, -CH_2_^a^-OH), 3.40 (1H, d, *J* = 16.0 Hz, H-8^b^), 3.34–3.25 (1H, m, -CH_2_^b^-OH), 3.06 (1H, dd, *J* = 10.9, 4.1 Hz, H-6^a^), 2.94 (1H, dddd, *J* = 13.3, 12.9, 6.5, 5.6 Hz, θ-CH_2_^a^), 2.87–2.77 (2H, m, H-5^a^, θ-CH_2_^b^), 2.56 (1H, d, *J* = 15.5 Hz, H-5^b^), 2.42 (1H, td, *J* = 11.4, 2.9 Hz, H-6^b^), 2.13 (1H, dd, *J* = 14.4, 9.6 Hz, C_13_-CH_2_^a^-CO), 1.95 (2H, h, *J* = 6.9 Hz, α-CH_2_), 1.81 (1H, dd, *J* = 14.4, 2.8 Hz, C_13_-CH_2_^b^-CO), 1.29–1.15 (6H, m, β-CH_2_, ζ-CH_2_, η-CH_2_), 1.08–1.01 (4H, m, δ-CH_2_, ε-CH_2_), 0.99–0.81 (2H, m, γ-CH_2_); ^13^C NMR (DMSO-*d*_6_, 101 MHz) δ (ppm) 172.08 (-CH-NH-CO-), 171.25 (-CH_2_-NH-CO-), 152.21 (NO_2_-Ph*_para_*), 150.05 (C-10), 146.21 (NO_2_-Ph*_ipso_*), 145.93 (C-2), 145.44 (C-3), 144.29 (C-9), 132.32 (C-12a), 129.16 (C-4a), 128.75 (C-13b), 127.54 (C-8a), 127.31 (2C, NO_2_-Ph*_meta_*), 124.05 (C-12), 122.73 (2C, NO_2_-Ph*_ortho_*), 110.73 (C-11), 108.03 (C-4), 105.81 (C-1), 100.63 (O-CH_2_-O), 69.28 (-CH-OH), 62.73 (C-13a), 60.59 (-CH_2_-OH), 59.43 (C_9_-OCH_3_), 55.70 (C_10_-OCH_3_), 55.60 (-NH-CH-), 53.90 (C-8), 50.59 (C-6), 39.48 (C-13), 38.89 (-NH-CO-CH_2_-C_13_), 38.45 (θ-CH_2_), 35.07 (α-CH_2_), 29.13 (C-5), 28.99 (η-CH_2_), 28.87 (δ-CH_2_), 28.69 (ε-CH_2_), 28.38 (γ-CH_2_), 26.45 (ζ-CH_2_), 25.32 (β-CH_2_).

*13-{[(8-{[(1R,2R)-1,3-dihydroxy-1-(4-nitrophenyl)propan-2-yl]carbamoyl}octyl)carbamoyl]methyl}-9,10-dimethoxy-5,6-dihydrobenzo[g]-1,3-benzodioxolo[5,6-a]quinolizinium hydroxide (CAM-C8-BER).* 97 mg (0.13 mmol) of CAM-C8-thBER and 46 mg (0.26 mmol) of NBS were dissolved in 5 mL of methylene chloride, and the resulting mixture was stirred for 1 h at RT. The volatiles were evaporated in vacuo. The target product was isolated on a silica gel column eluting with a solvent system of CHCl_3_:MeOH:NH_4_OH = 65:25:4. As a result, a yellow-brown solid was obtained. Yield: 57 mg (58%); TLC: *R_f_* (CHCl_3_:MeOH:NH_4_OH, 65:25:4) 0.47; LC-MS *m*/*z* calculated for C_40_H_47_N_4_O_10_ (M)^+^: 743.33, found 743.35; *t*_R_ = 1.62 min; ^1^H NMR (DMSO-*d*_6_, COSY, ROESY, 600 MHz) δ (ppm) 9.97 (1H, s, H-8), 8.77 (1H, t, *J* = 5.5 Hz, -CH_2_-NH-CO-), 8.19 (1H, d, *J* = 9.3 Hz, H-11), 8.13 (2H, d, *J* = 8.6 Hz, *o*-H NO_2_-Phe), 7.98 (1H, d, *J* = 9.3 Hz, H-12), 7.61 (1H, s, H-1), 7.59 (2H, d, *J* = 8.6 Hz, *m*-H NO_2_-Phe), 7.52 (1H, d, *J* = 9.2 Hz, -CH-NH-CO-), 7.15 (1H, s, H-4), 6.14 (2H, dd, *J* = 3.7, 1.1 Hz, O-CH_2_-O), 5.84 (1H, d, *J* = 5.5 Hz, -CH-OH), 5.03 (1H, dd, *J* = 5.5, 2.5 Hz, -CH-OH), 4.88 (1H, t, *J* = 5.4 Hz, -CH_2_-OH), 4.85 (2H, br s, H-6), 4.22 (2H, s, C_13_-CH_2_-CONH), 4.10 (3H, s, C_9_-OCH_3_), 4.06 (3H, s, C_10_-OCH_3_), 4.02 (1H, dddd, *J* = 9.2, 8.3, 5.2, 2.5 Hz, -CH-NH-CO-), 3.55 (1H, ddd, *J* = 10.4, 8.3, 6.4 Hz, -CH_2_^a^-OH), 3.29 (1H, dt, *J* = 10.4, 5.2 Hz, -CH_2_^b^-OH), 3.17 (2H, q, *J* = 6.5 Hz, θ-CH_2_), 3.10 (2H, t, *J* = 5.9 Hz, H-5), 2.05–1.87 (2H, m, α-CH_2_), 1.48 (2H, p, *J* = 7.1 Hz, η-CH_2_), 1.32–1.20 (4H, m, β-CH_2_, ζ-CH_2_), 1.20–1.06 (4H, m, δ-CH_2_, ε-CH_2_), 1.02–0.83 (2H, m, γ-CH_2_); ^13^C NMR (DMSO-*d*_6_, HMBC, HSQC, 151 MHz) δ (ppm) 172.07 (-CH-NH-CO-), 169.30 (-CH_2_-NH-CO-), 152.21 (NO_2_-Ph*_para_*), 150.30 (C-10), 149.30 (C-3), 146.67 (C-2), 146.17 (NO_2_-Ph*_ipso_*), 145.16 (C-8), 144.26 (C-9), 137.39 (C-13a), 133.99 (C-4a), 133.15 (C-12a), 128.05 (C-13), 127.34 (2C, NO_2_-Ph*_meta_*), 126.15 (C-11), 122.68 (2C, NO_2_-Ph*_ortho_*), 120.93 (C-8a), 120.82 (C-12), 120.16 (C-13b), 109.20 (C-1), 108.36 (C-4), 102.06 (O-CH_2_-O), 69.35 (-CH-OH), 62.09 (C_9_-OCH_3_), 60.53 (-CH_2_-OH), 57.04 (C_10_-OCH_3_), 56.87 (C-6), 55.69 (-NH-CH-), 38.99 (θ-CH_2_), 37.45 (-NH-CO-CH_2_-C_13_), 35.08 (α-CH_2_), 29.01 (η-CH_2_), 28.86 (δ-CH_2_), 28.66 (ε-CH_2_), 28.35 (γ-CH_2_), 27.24 (C-5), 26.52 (ζ-CH_2_), 25.31 (β-CH_2_).

### 3.6. Bacteria Inhibition Assays 

#### 3.6.1. Bacterial Strains

To prepare the bacterial suspension, bacterial stock cultures were sub-cultured onto Petri dishes containing 2.5% LB and 1.5% agar and incubated overnight at 37 °C until reaching the optical density of 1.5 (at 600 nm), which was measured on a Varioskan LUX microplate reader (Thermo Scientific, Waltham, MA, USA) or an Ultrospec 1100 pro spectrophotometer (Amersham Biosciences Corp., Piscataway, NJ, USA). A description of the strains is given in the Appendix A.

#### 3.6.2. Detection of Translation Inhibitors Using pDualrep2 Reporter Strain

For the in vitro bioactivity test, we used *Escherichia coli* reporter strain JW5503 (*ΔtolC*) (KanS) pDualrep2 as described previously [66,94]. Briefly, 1 µL of the solutions of CAM-Cn-BERs (20 mM), BER (20 mM), thBER-CH_2_-COOMe (20 mM), thBER-CH_2_-COOH (20 mM), BER-CH_2_-COOH (20 mM), CHL (2 mM), and LEV (70 nM) in DMSO were applied onto the agar plate that already contained a lawn of the reporter strain. After overnight incubation at 37 °C, the plate was scanned by ChemiDoc (Bio-Rad, Benicia, CA, USA) using “Cy3-blot” mode for RFP fluorescence and “Cy5-blot” mode for Katushka2S fluorescence.

#### 3.6.3. Testing the Antibacterial Activity of Substances on Plates with LB and Agar

*Escherichia coli* strains JW5503 (*ΔtolC*) (KanS) pCA24N*-LacZ,* and SQ110 (*ΔtolC)* (A2058G) were applied to Petri dishes filled with LB solid medium and agar. If necessary, selective antibiotics were added to the medium on plates. 1 µL of the tested compound at a concentration of 20 mM was applied to the surface of the dried dish; CHL (20 mM) and ERY (7 mM) were used as controls. After overnight incubation at 37 °C, the plates were scanned by ChemiDoc (Bio-Rad, Benicia, CA, USA) in three channels (Cy2, Cy3, and Cy5). The obtained images were processed in the Image Lab software (Bio-Rad).

#### 3.6.4. MIC Determination

The MICs for CAM-Cn-BERs and CH_3_-Cn-BERs were determined by Mueller-Hinton broth microdilution, as recommended by CLSI in the Methods for Dilution Antimicrobial Susceptibility Tests for Bacteria that Grow Aerobically, Approved Standard, 9th ed., CLSI document M07-A9, using in-house-prepared panels. The compounds were diluted in a 96-well microtiter plate to final concentrations ranging from 0.4 to 200 µM in a 200-µL aliquot of the bacterial suspension, followed by incubation at 37 °C for 18 h. The following strains of bacteria were used: *Escherichia coli* K-12, *Escherichia coli* JW5503 (*ΔtolC*) (KanS), *Escherichia coli* JW5503 (*ΔtolC)* (KanS) pCA24N*-LacZ*, *Bacillus subtilis* 168, *Bacillus subtilis* 168 pHT01. The MIC was determined as the lowest concentration that completely inhibited bacterial growth. The bacterial growth was observed visually alongside OD measurements. The experiments were carried out in triplicate.

#### 3.6.5. Assessment of Antibiotic Activity of Substances in Wells

Antimicrobial activity was determined by the agar diffusion method. To do this, we used a modified agar medium #2 Gause (%): glucose, 1.0; peptone, 0.5; tryptone, 0.3; NaCl, 0.5; agar, 2.0; tap water; pH 7.2–7.4. 100 µL aliquots of the solutions were loaded into wells 9 mm in diameter made in agar medium inoculated with test microorganisms. The following strains of microorganisms were used: Gram-positive bacteria *Bacillus subtilis* ATCC 6633, methicillin-resistant *Staphylococcus aureus* INA 00761 (MRSA),vancomycin-resistant *Leuconostoc mesenteroides* VKPM B-4177 (VRLM), *Mycobacterium smegmatis* VKPM Ac 1339, and *Mycobacterium smegmatis* mc^2^155; Gram-negative bacteria *Escherichia coli* ATCC 25922, *Escherichia coli* K-12; fungi *Aspergillus niger* INA 00760 and *Saccharomyces cerevisiae* RIA 259. The result was presented as the diameter (mm) of the zone where the growth of the test strain was suppressed.

### 3.7. In Vitro Translation Inhibition Assay 

The inhibition of firefly luciferase synthesis by the tested compounds was assessed in vitro, as described previously [95]. Briefly, the in vitro transcribed firefly luciferase mRNA (*Fluc*) was translated using the *E. coli* S30 Extract System for Linear Templates (Promega, Madison, WI, USA). Reaction mixtures containing 50 ng of mRNA and 0.1 µL of 10 mM of D-luciferin from the Steady-Glo Luciferase Assay System (Promega) were preincubated for 5 min with the tested compounds at a final concentration of 30 µM and carried out in 5-µL aliquots at 37 °C for 30 min. The activity of in vitro synthesized luciferase was measured by VICTOR X5 Multilabel Plate Reader (PerkinElmer, Waltham, MA, USA) every 30 s. 

The inhibition of eukaryotic translation was measured in a lysate of HEK293T cells (S10 extract) according to the protocol given in the Appendix A. 

### 3.8. In Vitro Binding Assay 

Binding affinities of CAM-Cn-BERs, as well as other tested compounds, for the *E. coli* ribosome, were analyzed by a competition-binding assay using fluorescently labeled BODIPY-ERY as described before [22,73,74]. BODIPY-ERY (16 nM) was incubated with ribosomes (47 nM) for 30 min at 25 °C in the buffer containing 20 mM HEPES-KOH (pH 7.5), 50 mM NH_4_Cl, 10 mM Mg(CH_3_COO)_2_, 4 mM β-mercaptoethanol, and 0.05% Tween-20. Solutions of CHL, CAM-Cn-BERs, CH_3_-Cn-BERs were added to ribosome/BODIPY-ERY complexes to final concentrations ranging from 0.01 to 200 µM. The mixtures were incubated for 2 h at room temperature until equilibrium was reached, and then the values of fluorescence anisotropy were measured with VICTOR X5 Multilabel Plate Reader (PerkinElmer, Waltham, MA, USA) using a 384-well plate. The excitation wavelength was 485 nm, and the emission wavelength was 530 nm. The apparent dissociation constants were calculated for each tested compound based on the assumption that the competitive equilibrium binding of two ligands occurs at a single binding site, as described in [96].

### 3.9. Toeprinting Analysis 

Assessment of the inhibition of primer extension (toeprinting) by the tested compounds was carried out using the *RST-2* template as previously described [97]. The final concentrations of the tested compounds were 30 µM.

The sequence of the used matrix *RST-2* in the direction 5′→3′: ACTAATACGACTCACTATAGGGCTTAAGTATAAGGAGGAAAACATATGAAATTCGCCATCACCCTGCGTCAGTGCGAAGGCTGGTCACCGGTACATTATGACAATTAATAATAATAAAAAAAGTGATAGAATTCTATCGTTAATAAGCAAAATTCATTATAACC. For reverse transcription NV1 oligonucleotide was used, its sequence: GGTTATAATGAATTTTGCTTATTAAC.

### 3.10. Molecular Docking

Molecular docking was performed using the AutoDock Vina 1.2.3 software [98,99]. For docking, the structure of the *T. thermophilus* ribosome in complex with aminoacylated A-site Phe-NH-tRNA^Phe^, peptidyl P-site fMTHSMRC-NH-tRNA^Met^, and deacylated E-site tRNA^Phe^ (PDB ID: 8CVL [76]) was used, from which water molecules and inorganic ions were removed. The A-site tRNA was deacylated, and the fMTHSMRC peptide at the P-site tRNA^Met^ in the initial crystal structure was replaced by the fMKFAI-pentapeptide using PyMol 2.5 (www.pymol.org, accessed on 20 October 2022) and Avogadro [100] software. The initial geometry of CAM-Cn-BERs molecules was generated by a semi-empirical method in the MOPAC2016 software using the PM7 Hamiltonian [101]. Polar hydrogen atoms were added to the structures of the ribosome, and studied molecules and charges were assigned by Autodock Tools 1.5.7 software. The 50S ribosomal subunit in complex with deacylated A-site tRNA^Phe^ and P-site tRNA^Met^ was considered as the rigid part of the receptor, while the fMKFAI peptide was flexible but fixed to the P-site tRNA^Met^ with its C-terminus during calculations. The molecular docking area was defined as a cuboid, which included the CHL binding site in the PTC and the upper part of the NPET. Vinardo scoring function was used for calculations [102]. For each compound, 20 conformations were obtained, which were ranked according to the calculated energy of interaction with the ribosome. File formats were converted by Open Babel 3.1.1 software [103]. The figures were generated using PyMol 2.5 software.

### 3.11. Measurement of B. subtilis Membrane Potential 

The membrane potential of *B. subtilis* was estimated by measuring the fluorescence of the potential-dependent probe, diS-C_3_-(5) [104]. *B. subtilis* from the overnight culture were seeded into a fresh LB medium, followed by growth for 24 h until reaching the optical density of 0.8 at 600 nm. Then the bacteria were diluted 20-fold in a buffer containing 100 mM of KCl and 10 mM of Tris, pH 7.4. The fluorescence was measured at 670 nm (excitation at 630 nm) using a Fluorat-02-Panorama fluorimeter (Lumex Instruments, St.Petersburg, Russia).

### 3.12. Biofilm Development

To assess how compounds affect biofilm formation, we modified the protocol [105] and evaluated the effects of a substance on biofilm formation as a change in the ratio of planktonic and sessile forms of bacteria. 

The microbial biofilms were cultivated in LB media in polystyrene 96-well plates (Citotest, Haimen, China). Panels of test substances were prepared by the method of double dilutions, 200 µL of which was added to each well. Antibiotics were not added to control samples. *E. coli* K12 cell suspension (5 × 10^5^ cells per mL) was added to each well. Microtiter plates were incubated at 37 °C in a Thermo Scientific Multiskan FC plate reader with an incubator (Thermo Fisher Scientific, Waltham, MA, USA) for 20 h. Bacterial growth was observed by means of OD_620_ measurements each hour [106]. After incubation, the liquid media was discarded by inverting the plate upside down to dump the cell suspension, and the plate was triply washed with PBS buffer to remove loose cells. To fix the remaining biofilms, the plates were placed in a thermostat at 60 °C for 30 min. To stain biofilms, 40 µL of 1% crystal violet was added to each well and left to stain for 30–60 min. The crystal violet was discarded, and the plate was washed twice with DI water, and then 200 mL of 95% ethanol was added into each well to extract the crystal violet. Measurement of the absorbance of crystal violet was performed at 595 nm using the microplate reader. All biofilm samples were prepared in quadruplicate.

### 3.13. In Vitro Survival Assay (MTT Assay)

The cytotoxicity of the substances under study was tested using the MTT (3-(4,5-dimethylthiazol-2-yl)-2,5-diphenyltetrazolium bromide) assay [88], with some modifications. Two thousand five hundred cells per well for the MCF7, HEK293T, and A549 cell lines or 4000 cells per well for the VA13 cell line were plated in 135 µL of DMEM-F12 media with 10% FBS (both Gibco, Waltham, MA, USA) in a 96-well plate and incubated in a 5% CO_2_ incubator for the first 16 h, without treatment. Then 15 µL of media-DMSO solutions of the tested substances were added to the cells (the final DMSO concentrations in the media were 1% or less), and the cells were treated for 72 h with 25 nM–50 µM (eight dilutions) of our substances (triplicate each). Serial dilutions of DMSO and doxorubicin were used as controls. The MTT reagent (Paneco LLC, Moscow, Russia) was then added to the cells to a final concentration of 0.5 g/L (10× stock solution in PBS was used) and incubated for 2.5 h at 37 °C in the incubator under an atmosphere of 5% CO_2_. The MTT solution was then discarded, and 140 µL of DMSO (PharmaMed LLC, Krasnodar, Russia) was added. The plates were swayed on a shaker (60 rpm) to dissolve the formazan. The absorbance was measured using VICTOR X5 Multilabel Plate Reader (PerkinElmer, Waltham, MA, USA) at a wavelength of 565 nm (in order to measure formazan concentration). The results were used to construct a dose-response graph and to estimate IC50 values.

## 4. Conclusions

Conjugates of chloramphenicol amine (CAM) and berberine (BER) via alkylamide linkers of different lengths, introduced in the C-13 position of BER, CAM-Cn-BER (n = 1, 2, 3, 5, 8), were synthesized in this study as potential antimicrobial compounds combining two different biologically active fragments within a single chimeric molecule. Taking into account knowledge of the molecular targets of the parent compounds, chloramphenicol (CHL) and BER, we examined the ribosome binding and translation inhibitory properties of new conjugates, as well as their effects on the bacterial membrane. Similarly to the parent CHL, berberine analogs of CHL, CAM-Cn-BERs, strongly bind to the 70S ribosome and specifically inhibit protein synthesis in vitro, allowing the formation of short peptides. At the same time, an analog with a relatively long alkyl linker—CAM-C8-BER—decreases the membrane potential of bacterial cells similarly to the 13-nonylamide derivative of berberine (CH_3_-C8-BER). Moreover, CAM-C8-BER suppresses some laboratory strains of CHL and macrolides-resistant bacteria, acts against mycobacteria, and has a narrower spectrum of action on Gram-positive bacteria than CHL and 13-substituted nonylamide derivative of BER, acting against a vancomycin-resistant strain *L. mesenteroides*. Thus, the action of CAM-C8-BER can have a double effect on bacterial cells associated with the inhibition of protein biosynthesis and depolarization of the bacterial membrane, similar to the mechanism of action described earlier in the triphenylphosphonium analogs of CHL [23], which are conceptually similar to CAM-Cn-BERs. Despite the high affinity for the ribosome and improved antibacterial properties compared to CHL, the main drawback of the triphenylphosphonium analogs of CHL was their toxicity. A significant advantage of this work in comparison with previous studies is the fact that CAM-Cn-BERs having antibacterial properties, are not toxic to eukaryotic cells. Moreover, an important property of CAM-C8-BER is its ability not to induce biofilm formation, unlike the parent CHL.

## Figures and Tables

**Figure 1 antibiotics-12-00015-f001:**
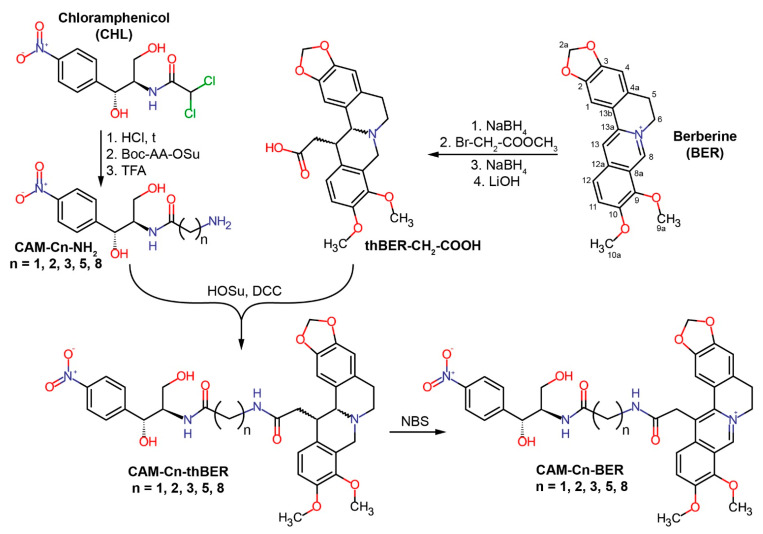
Scheme of the chemical synthesis of conjugates of chloramphenicol amine (CAM) and berberine (BER) containing linkers of different lengths—CAM-Cn-BER, n = 1, 2, 3, 5, 8.

**Figure 2 antibiotics-12-00015-f002:**
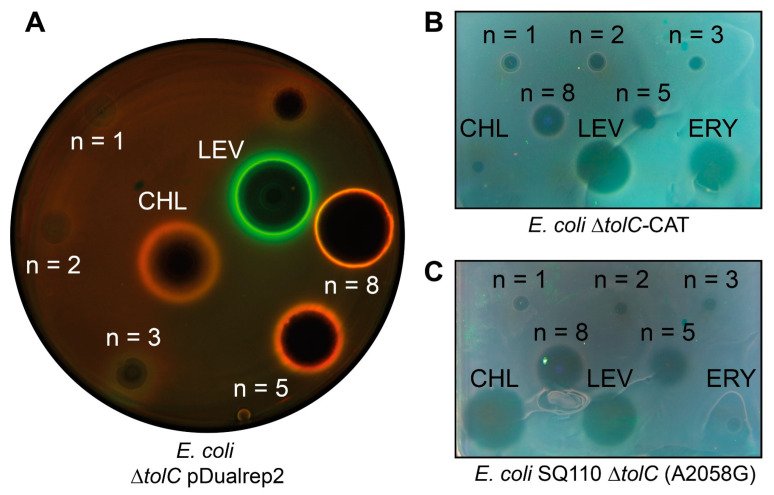
Antibacterial activity of CAM-Cn-BER, n = 1, 2, 3, 5, 8. Levofloxacin (LEV), chloramphenicol (CHL), and erythromycin (ERY) are used as the controls. (**A**) Testing of the CAM-Cn-BER activity using *E. coli*
*ΔtolC* pDualrep2 reporter strain. The induction of the red fluorescent protein expression (green halo around the inhibition zone, pseudocolor) is triggered by DNA damage, while the induction of Katushka2S protein (red halo, pseudocolor) occurs in response to ribosome stalling. The unlabeled spot above LEV corresponds to a substance not studied in this work. (**B**) Testing of the CAM-Cn-BER antibacterial activity on CHL-resistant *E. coli* strain *ΔtolC*-CAT harboring pCA24N*-LacZ* plasmid with the *cat* gene encoding for chloramphenicol acetyltransferase. (**C**) Testing of the CAM-Cn-BER antibacterial activity on macrolide-resistant *E. coli* strain SQ110 *ΔtolC* (A2058G) in which the A2058 nucleotide is replaced by G2058 in the 23S rRNA.

**Figure 3 antibiotics-12-00015-f003:**
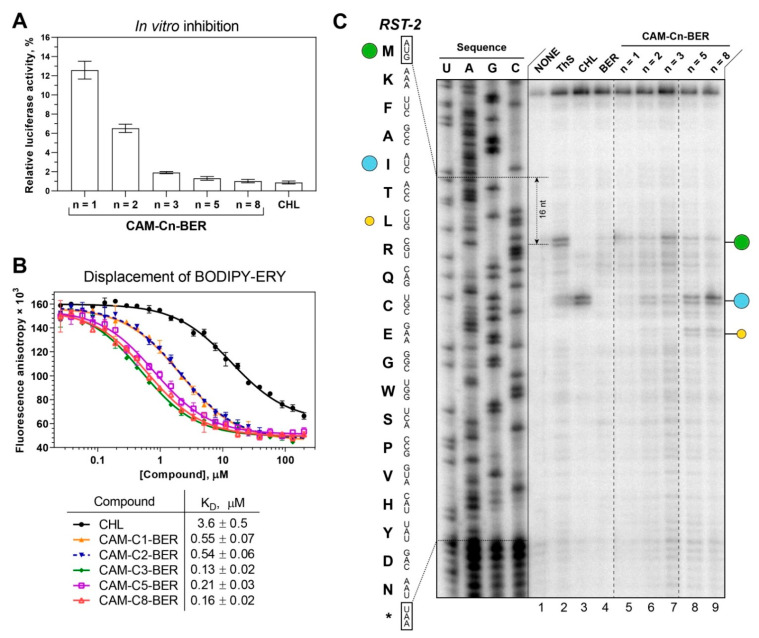
Binding affinity to bacterial ribosomes and the inhibition of protein synthesis by CAM-Cn-BER, n = 1, 2, 3, 5, 8. (**A**) The inhibition of protein synthesis in vitro by 30 µM of CHL and CAM-Cn-BERs in the cell-free bacterial transcription-translation coupled system. The relative enzymatic activity of in vitro synthesized firefly luciferase is shown. The error bars represent the standard deviations. (**B**) A competitive binding assay to test the affinity of CHL and CAM-Cn-BERs to *E. coli* 70S ribosomes measured by fluorescence anisotropy of fluorescently labeled analog of the erythromycin, BODIPY-ERY. All reactions were repeated at least two times. Error bars represent the standard deviation. The resulting mean values for the apparent dissociation constants (K_Dapp_) with confidence intervals (α = 0.05) are shown in the table below the graph. (**C**) Ribosome stalling by CAM-Cn-BER (n = 1, 2, 3, 5, 8, lanes 5–9) on *RST-2* mRNA as detected by a reverse-transcription primer-extension inhibition (toeprinting) assay in a cell-free translation system. DMSO, 0.5% (NONE, lane 1), ThS (inhibits translation at the start codon, lane 2), CHL (lane 3), and BER (lane 4) were used as controls. The nucleotide sequence of *RST-2* mRNA and its corresponding amino acid sequence are shown on the left. The green circle marks the translation arrest at the start codon, while the blue and yellow circles denote drug-induced arrest sites within the coding sequence of the mRNA used. Note that due to the large size of the ribosome, the reverse transcriptase used in the toeprinting assay stops 16 nucleotides downstream of the codon located in the P-site. The asterisk (*) indicates a stop codon. Dashed lines mark the places from which parts of the gel were cut out. The full image of the gel is shown in Appendix A.

**Figure 4 antibiotics-12-00015-f004:**
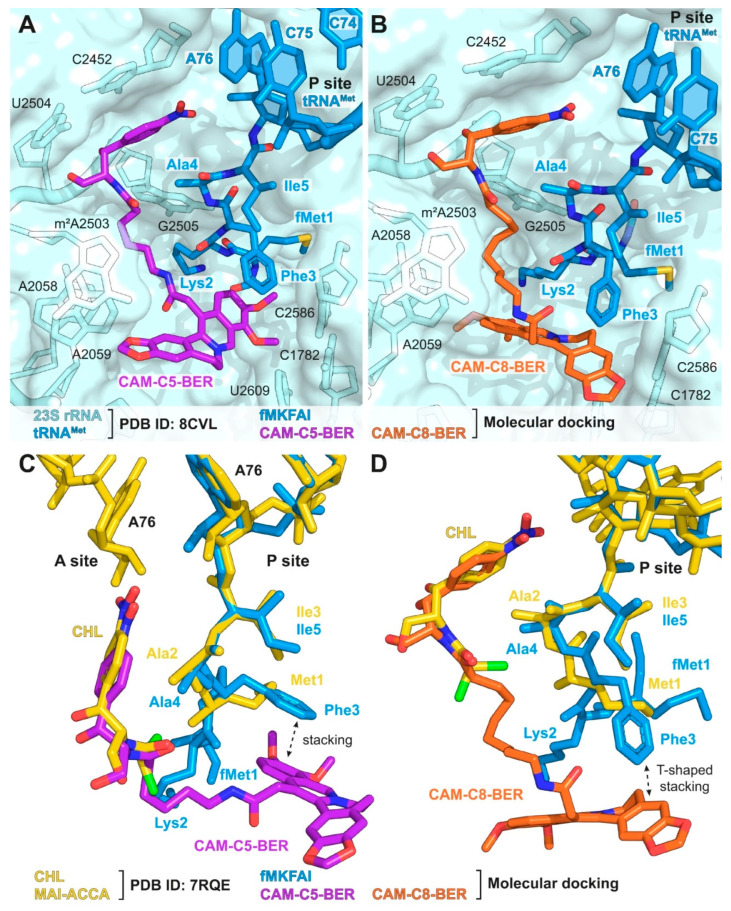
Structures of complexes of CAM-C5-BER and CAM-C8-BER with the *T. thermophilus* ribosome obtained by molecular docking. (**A,B**) Structures of CAM-C5-BER ((**A**), purple), CAM-C8-BER ((**B**), orange), and fMKFAI-NH-tRNA^Met^ (blue) docked into the crystal structure of *T. thermophilus* ribosome (PDB ID: 8CVL [76], light blue). (**C,D**) Superposition of the structures of CAM-C5-BER ((**C**), purple) and CAM-C8-BER ((**D**), orange) in complex with *T. thermophilus* ribosome carrying fMKFAI-NH-tRNA^Met^ (blue) obtained by molecular docking with the previously reported crystal structure of CHL (yellow) and P-site MAI-ACCA (yellow) in complex with *T. thermophilus* ribosome (PDB ID: 7RQE [16]). *E. coli* numbering of nucleotides is presented.

**Figure 5 antibiotics-12-00015-f005:**
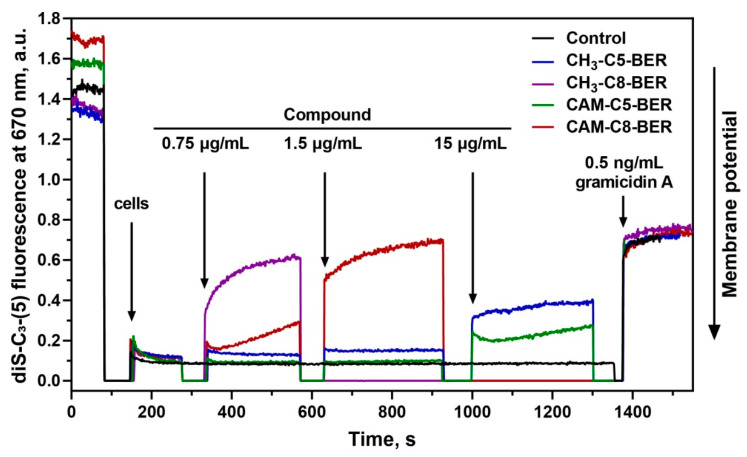
Effect of CAM-Cn-BER (n = 5, 8) and CH_3_-Cn-BER (n = 5, 8) on the kinetics of the membrane potential of *B. subtilis* cells measured using the fluorescent probe diS-C_3_-(5). Arrows mark moments at which appropriate amounts of the compounds were added. Gramicidin A at a concentration of 0.5 ng/mL was used as a control.

**Figure 6 antibiotics-12-00015-f006:**
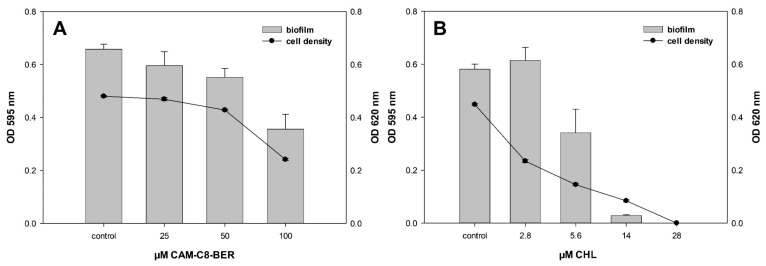
Biofilm formation of *E. coli* at different concentrations of CAM-C8-BER (**A**) and CHL (**B**). *E. coli* WT cells were exposed 20 h with the test compounds, and their cell density (black curves) and surface attachment (gray bars) were measured. Values for cell density (OD_620_) are indicated on the right y-axis, and biofilm values (OD_595_) are indicated on the left y-axis.

**Table 1 antibiotics-12-00015-t001:** Suppression of the growth of *E. coli* and *B. subtilis* strains, including CHL-resistant ones, by CAM-C8-BER. CH_3_-C8-BER and CHL were used as controls. The values of a minimal inhibitory concentration (MIC, µM) are shown ^1^.

	*E. coli* K12	*E. coli* JW5503 *ΔtolC*	*E. coli* JW5503 *ΔtolC*-CAT	*B. subtilis* 168	*B. subtilis*-CAT 168
CAM-C8-BER	>100	12.5	50	100	50
CH_3_-C8-BER	>100	12.5	12.5	12.5	12.5
CHL	6	2.5	>100	12.5	>100

^1^ The MIC values were determined using the double-dilution method. The MIC for each compound was determined in triplicate in two independent sets.

**Table 2 antibiotics-12-00015-t002:** Growth inhibition of various Gram-positive and Gram-negative strains of bacteria, as well as fungi and yeasts. The values of diameters of zones of growth inhibition of the studied strains (mm) are shown ^1^. Antimicrobial activity was determined by the agar diffusion method. 100 µL aliquots of the solutions (0.2 mg/mL) were loaded into wells 9 mm in diameter made in agar medium inoculated with test microorganisms.

	*B. subtilis* ATCC 6633	*L. mesenteroides*VKPM B-4177	*St. aureus* INA 00761	*Myc. smegmatis*VKPM Ac 1339	*Myc. smegmatis* mc^2^155	*E. coli* ATCC 25922	*E. coli* K-12	*A. niger* INA 00760	*S. cerevisiae* RIA 259
CAM-C8-BER	NO ^2^	11(13)	NO	13	15	NO	NO	NO	NO
CH_3_-C8-BER	15	18	20	14	22	NO	NO	16	12
BER	NO	NO	NO	14	12	NO	NO	NO	NO
CHL	30	25	23	11	21	18	26	NO	NO

^1^ Zones of partial growth inhibition are indicated in brackets. ^2^ NO (not observed)—no inhibition.

**Table 3 antibiotics-12-00015-t003:** Growth inhibition by CAM-Cn-BER in relation to a number of cell lines according to the MTT assay. Values of a half-maximal inhibitory concentration (IC50, µM) are shown.

	HEK293T	MCF7	A549	VA13
CHL	>50	>50	>50	>50
CAM-C5-BER	>50	>50	>50	>50
CAM-C8-BER	28 ± 3	>50	>50	>50
CH_3_-C5-BER	11 ± 1	>50	34 ± 6	>50
CH_3_-C8-BER	1.39 ± 0.07	1.8 ± 0.2	1.7 ± 0.1	2.9 ± 0.3
BER	0.70 ± 0.08	25 ± 4	4.6 ± 0.5	17 ± 2
Doxorubicin	0.007 ± 0.001	0.04 ± 0.01	0.04 ± 0.01	0.18 ± 0.04

## Data Availability

All data are presented in the publication and Appendix A.

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
