# Peer review of "Conjugates of Chloramphenicol Amine and Berberine as Antimicrobial Agents"

_antibiotics, 2022, doi:10.3390/antibiotics12010015_

Round 1
Reviewer 1 Report
Research article by Julia A. Pavlova et al, showed evidence of the antimicrobial potential of synthesized conjugates of chloramphenicol amine and berberine. Showed binding of synthesized compounds to the bacterial ribosome, inhibiting protein synthesis and showed these compounds do not induce biofilm using biochemical assays.
The manuscript writing is good however there are certain changes that need to be made. I would like to accept the manuscript with minor changes as the review article is useful and interesting.
Below are suggestions for minor revision of the manuscript:
1. Abstract section mention they have shown CAM- cn -BER compounds bind to ribosome and inhibit protein synthesis both in vitro and in vivo but the authors have not shown the In vivo data.
2. It would be nice to have an additional experiment showing antimicrobial activity in eukaryotic cells by infecting with bacterial pathogens and treating with newly synthesized derivative compounds (CAM- cn -BER compounds).
3. Biofilm assays images would have been documented in the manuscript but not mandatory.
I have no major concerns with this paper to publish in antibiotics Journal after minor revision.
Author Response
We are grateful to the Reviewers for the careful reading of our manuscript, as well as the valuable comments made.
Below we present the Authors' responses to the Reviewers' comments.
Reviewer 1 comments
- Abstract section mention they have shown CAM- cn -BER compounds bind to ribosome and inhibit protein synthesis both in vitro and in vivo but the authors have not shown theIn vivo data.
Response: The abstract to the article has been rewritten more carefully.
- It would be nice to have an additional experiment showing antimicrobial activity in eukaryotic cells by infecting with bacterial pathogens and treating with newly synthesized derivative compounds (CAM- cn -BER compounds).
Response: In our work, we presented the results of the action of the synthesized compounds (CAM-Cn-BER) on 4 eukaryotic strains (MTT test, Table 3), and also tested the effect of these compounds on the process of eukaryotic translation in the in vitro system. In addition, we showed that the compounds did not inhibit the growth of A. niger and S. cerevisiae strains (Table 2). Since we consider the synthesized compounds to be quite promising, we will continue their research in the future, including on eukaryotic cells.
- Biofilm assays images would have been documented in the manuscript but not mandatory.
Response: When analyzing biofilms, we comprehensively analyzed the data obtained. First, we measured both growth kinetics and endpoint absorbance, which rules out an anomalous data distribution. Biofilms were assessed visually, and then the data were compared with the data on crystal violet staining in the wells. We have never seen any significant discrepancies in the results. These were true both for substances that do not affect or enhance the growth of the biofilm, and for substances that suppress it. Similarly, the extraction of crystal violet did not lead to abnormal results and was proportional to the initial level of biofilms. In this regard, we did not conduct any special documentation of biofilms, since the experiment was usually accompanied by negative and positive controls. But we agree with the Reviewer that the inclusion of such data would probably be a useful, albeit expected, result, and we will definitely take this into account in our future research and articles.
Reviewer 2 Report
Pavlova et al. report the effects of new molecules that consist of chloramphenicol and berberine linked with an aliphatic chain of different length. These new compounds exert effects that are similar to the starting compounds, showing that the linkage does not affect either activity. The study is essentially a continuation of previous work linking other moieties to chloramphenicol. The experimental approach and methods used are therefore well established and are well described. These are intriguing results with potential implications for antibiotic drug development.
The manuscript is generally well written with some minor English language weaknesses.
Items to be addressed:
Fig. 2a: There is an additional spot visible at the top above LEV that is not labeled. This should be labeled or explained.
Table 1: it seems that the CH3 control has a lower MIC than the active compound. Why is that? Are the labels correct? If correct, this was not noted in the text.
Line 681: Reference 95 is given for the luciferase translation assay but ref 95 is a crystal structure study that does not seem to use this assay (at least on a quick glance).
English language issues:
Line 54: ‘to develop the design of new….’ Might be better if only one was used, either to develop or to design.
Line 117: ‘relatively large doses require for its action’
Line 155: better to use ‘did not’ instead of ‘didn’t’
Line 409: ‘forms a stacking with them’
Author Response
We are grateful to the Reviewer for the careful reading of our manuscript, as well as the valuable comments made.
Below we present the Authors' responses to the Reviewers' comments.
Fig. 2a: There is an additional spot visible at the top above LEV that is not labeled. This should be labeled or explained.
Response: The spot above LEV corresponds to a substance not included in this work. The corresponding explanation is added to the caption of Fig. 2.
Table 1: it seems that the CH3 control has a lower MIC than the active compound. Why is that? Are the labels correct? If correct, this was not noted in the text.
Response: Yes, this is really an experimental fact. We write in the text: « Notably, the tested substances are active against CHL-resistant strains E. coli ∆tolC-CAT and B. subtilis-CAT, indicating that they can overcome resistance induced by the activity of chloramphenicol acetyltransferases. However, the control substance CH3-C8-BER also shows significant antibacterial activity against both resistant and non-resistant strains. Apparently, the berberine fragment in the conjugate mainly contributes to the effect of CAM-C8-BER on resistant strains.»
Line 681: Reference 95 is given for the luciferase translation assay but ref 95 is a crystal structure study that does not seem to use this assay (at least on a quick glance).
Response: We agree with the comment, reference 95 was incorrect and has been replaced by “Lukianov, D.A.; Buev, V.S.; Ivanenkov, Y.A.; Kartsev, V.G.; Skvortsov, D.A.; Osterman, I.A.; Sergiev, P.V. Imidazole Derivative As a Novel Translation Inhibitor. Acta Naturae 2022, 14(2), 71–77. doi: 10.32607/actanaturae.11654”. We additionally checked all other references and removed ref. 96.
English language issues (line 54, 117, 155, 409).
Response: Corrected.
Reviewer 3 Report
Congratulation on a well-designed study. The search for novel strategies for microbial control is much relevant and the rationale for the selected precursors is clearly defined. Some minor considerations to improve the quality of the work.
General: There are sections with no space between the titles and the text or some titles in all caps. These revise the manuscript for format.
Abstract
The abstract is lacking a brief background before pursuing the main goal of the work. Also, the text is somewhat vague as a result of the large among work to report and the inclusion of more specific and detailed results may be relevant.
Introduction
This section is too extensive. The authors provided a deepened overview of the available bibliography but some of the reflections included in this section may be better suited for the discussion of some results.
Lines 138-157 – There are some results included in this section. This overall summary of the results should be relocated to the conclusions section and removed from the introduction.
Results and discussion
Section: CAM-Cn-BER and CH3-Cn-BER Exhibit Antibacterial Activity Against Various Strains, Including Resistant
In table 2, results of no inhibition should be presented as No inhibition (NO – not observed) or other abbreviation and not with a zero mm, as it is difficult the reading the table.
Section 2.10
MTT is a marker for metabolic activity and not for cell growth or proliferation as these two concepts are not directly transposable. Please adopt the IC50 (Half-maximal inhibitory concentration) term in replace of GI50)
Methods
3.6.1 bacterial strains
Please described the used culture medium
3.11 and other sections
Some references are italicized.
Author Response
We are grateful to the Reviewer for the careful reading of our manuscript, as well as the valuable comments made.
Below we present the Authors' responses to the Reviewers' comments.
General: There are sections with no space between the titles and the text or some titles in all caps. These revise the manuscript for format.
Response: We have edited and formatted the text according to the rules of the journal.
Abstract
The abstract is lacking a brief background before pursuing the main goal of the work. Also, the text is somewhat vague as a result of the large among work to report and the inclusion of more specific and detailed results may be relevant.
Response: The abstract to the article has been rewritten more carefully.
Introduction
This section is too extensive. The authors provided a deepened overview of the available bibliography but some of the reflections included in this section may be better suited for the discussion of some results.
Response: We have slightly shortened the Introduction section. In addition, in the Results and Discussion section, we have added references to the works cited in the Introduction, and an explonatory text.
Lines 138-157 – There are some results included in this section. This overall summary of the results should be relocated to the conclusions section and removed from the introduction.
Response: We have removed the text describing the results from the Introduction. Part of this text has been moved to the Abstract.
Results and discussion
Section: CAM-Cn-BER and CH3-Cn-BER Exhibit Antibacterial Activity Against Various Strains, Including Resistant
In table 2, results of no inhibition should be presented as No inhibition (NO – not observed) or other abbreviation and not with a zero mm, as it is difficult the reading the table.
Response: Corrected.
Section 2.10
MTT is a marker for metabolic activity and not for cell growth or proliferation as these two concepts are not directly transposable. Please adopt the IC50 (Half-maximal inhibitory concentration) term in replace of GI50)
Response: We agree with the Reviewer's comment and have made the necessary changes to the Table 3.
Methods
3.6.1 bacterial strains
Please described the used culture medium
Response: We have added the composition of the culture medium (containing 2.5% LB and 1,5% agar) to the description of the experiment in section 3.6.1.
3.11 and other sections
Some references are italicized
Response: Corrected.
In addition to the corrections in the text recommended by the Reviewers, we also made some edits related to some overlaps in the content with the published articles. However, I would like to note that overlaps in the text in an experimental article occur for the following reasons. In the case when published data are described (with references to relevant articles), it is logical to use the words of the Authors to cite their results, especially when it comes to our own previously obtained results. Most of the overlaps in the text relate to experimental terms, names and procedures that can only be used in this way and in no other way.